# Forecasting in Blockchain-Based Local Energy Markets

**Michael Kostmann [1,\*]**  **and Wolfgang K. Härdle [2,3,4]**

1   School of Business and Economics, Humboldt-Universität zu Berlin, Spandauer Str. 1, 10178 Berlin, Germany
2   Ladislaus von Bortkiewicz Chair of Statistics, School of Business and Economics,
    Humboldt-Universität zu Berlin, Unter den Linden 6, 10099 Berlin, Germany
3   Wang Yanan Institute for Studies in Economics, Xiamen University, 422 Siming Road, Xiamen 361005, China
4   Department of Mathematics and Physics, Charles University Prague, Ke Karlovu 2027/3,
    12116 Praha 2, Czech
\*   Correspondence: michael.kostmann@hu-berlin.de

**Abstract:** Increasingly volatile and distributed energy production challenges traditional mechanisms to manage grid loads and price energy. Local energy markets (LEMs) may be a response to those challenges as they can balance energy production and consumption locally and may lower energy costs for consumers. Blockchain-based LEMs provide a decentralized market to local energy consumer and prosumers. They implement a market mechanism in the form of a smart contract without the need for a central authority coordinating the market. Recently proposed blockchain-based LEMs use auction designs to match future demand and supply. Thus, such blockchain-based LEMs rely on accurate short-term forecasts of individual households' energy consumption and production. Often, such accurate forecasts are simply assumed to be given. The present research tested this assumption by first evaluating the forecast accuracy achievable with state-of-the-art energy forecasting techniques for individual households and then, assessing the effect of prediction errors on market outcomes in three different supply scenarios. The evaluation showed that, although a LASSO regression model is capable of achieving reasonably low forecasting errors, the costly settlement of prediction errors can offset and even surpass the savings brought to consumers by a blockchain-based LEM. This shows that, due to prediction errors, participation in LEMs may be uneconomical for consumers, and thus, has to be taken into consideration for pricing mechanisms in blockchain-based LEMs.

**Keywords:** blockchain; local energy market; smart contract; smart meter; short-term energy forecasting; machine learning; least absolute shrinkage and selection operator (LASSO); long short-term memory (LSTM); prediction errors; market mechanism; market simulation

**JEL Classification:** Q47; D44; D47; C53

## 1. Introduction

The "Energiewende", or energy transition, is a radical transformation of Germany's energy sector towards carbon free energy production. This energy revolution led in recent years to widespread installation of renewable energy generators [1,2]. In 2017, more than 1.6 million photovoltaic micro-generation units were already installed in Germany [3]. Although this is a substantial step towards carbon free energy production, there is a downside: The increasing amount of distributed and volatile renewable energy resources, possibly combined with volatile energy consumption, presents a serious challenge for grid operators. As energy production and consumption have to be balanced in electricity grids at all times [4], modern technological solutions to manage grid loads and price renewable energy are needed. One possibility to increase the level of energy distribution efficiency

on low aggregation levels is the implementation of local energy markets (LEMs) in a decentralized approach, an example being the Brooklyn Microgrid [5].

LEMs enable interconnected energy consumers, producers, and prosumers to trade energy in near real-time on a market platform with a specific pricing mechanism [6]. A common pricing mechanism used for this purpose are discrete double auctions [7–9]. Blockchain-based LEMs utilize a blockchain as underlying information and communication technology and a smart contract to match future supply and demand and to settle transactions [10]. As a consequence, a central authority that coordinates the market is obsolete in a blockchain-based LEM. Major advantages of such LEMs are the balancing of energy production and consumption in local grids [11], lower energy costs for consumers [12], more customer choice (empowerment) [13], and less power line loss due to shorter transmission distances [14].

In the currently existing energy ecosystem, the only agents involved in electricity markets are utilities and large-scale energy producers and consumers. Household-level consumers and prosumers do not actively trade in electricity markets. Instead, they pay for their energy consumption or they are reimbursed for their infeed of energy into the grid according to fixed tariffs. In LEMs, on the contrary, households are the participating market agents that typically submit offers in an auction [7,15]. This market design requires the participating households to estimate their future energy demand and/or supply, to be able to submit a buy or sell offer to the market [16]. Therefore, accurate forecasts of household energy consumption/production are a necessity for such LEM designs. This is due to the market mechanism employed and does not depend on whether an LEM is implemented on a blockchain or not. However, research on blockchain-based LEM mostly employ market mechanisms that require accurate forecasts of household energy consumption/production making the aspect of forecasting especially relevant here. Despite this, it is frequently assumed in existing research on (blockchain-based) LEMs that such accurate forecasts are readily available (see, e.g., [6–8,16,17]). However, forecasting the consumption/production of single households is difficult due to the inherently high degree of uncertainty, which cannot be reduced by the aggregation of households [18]. Hence, the assumption that accurate forecasts are available cannot be taken in practice to be correct. Additionally, given the substantial uncertainty in individual households' energy consumption or production, prediction errors may have a significant impact on market outcomes.

This is where we focused our research: We evaluated the possibility of providing accurate short-term household-level energy forecasts with existing methods and currently available smart meter data. Moreover, our study aimed to quantify the effect of prediction errors on market outcomes in blockchain-based LEMs. For the future advancement of the field, it seemed imperative that the precondition of accurate forecasts of individual households' energy consumption and production for LEMs is assessed. Because, if the assumption cannot be met, the proposed blockchain-based LEMs may not be a sensible solution to support the transformation of our energy landscape. This, however, is urgently necessary to limit $CO_2$ emissions and the substantial risks of climate change.

## 1.1. Related Research

Although LEMs started to attract interest in academia already in the early 2000s, it is still an emerging field [11]. Mainly driven by the widespread adoption of smart meters and Internet-connected home appliances, recent work on LEMs focuses on use cases in developed and highly technologized energy grid systems [19]. While substantial work regarding LEMs in general has been done (e.g., [7,8,15]), there are only few examples of blockchain-based LEM designs in the existing literature. Mengelkamp et al. [10] derived seven principles for microgrid energy markets and evaluated the Brooklyn Microgrid according to those principles. With a more practical focus, Mengelkamp et al. [6] implemented and simulated a local energy market on a private Ethereum-blockchain that enables participants to trade local energy production on a decentralized market platform with no need for a central authority. Münsing et al. [20] similarly elaborate a peer-to-peer energy market concept on a blockchain but focus on operational grid constraints and a fair payment rendering. Additionally,

there are several industry undertakings to put blockchain-based energy trading into practice, such as Grid Singularity (gridsingularity.com) in Austria, Powerpeers (powerpeers.nl) in the Netherlands, Power Ledger (powerledger.io) in Australia, and LO3 Energy (lo3energy.com) in the United States.

Interestingly, none of the above cited works, that employ market mechanisms requiring household energy forecasts for bidding, check whether the assumed availability of such forecasts is given. However, without this assumption, trading through an auction design, as described by, e.g., Block et al. [9] or Buchmann et al. [8], and implemented in a smart contract by Mengelkamp et al. [6] is not possible. Unfortunately, this forecasting task is not trivial due to the extremely high volatility of individual households' energy patterns [18]. However, research by Arora and Taylor [21], Kong et al. [22], Shi et al. [23], and Li et al. [24] shows that advances in the energy forecasting field also extend to household-level energy forecasting problems and serve as a promising basis for the present study.

*1.2. Present Research*

We investigated the prerequisites necessary to implement blockchain-based distributed local energy markets. In particular this means:

(a)   forecasting net energy consumption and production of private consumers and prosumers one time-step ahead;
(b)   evaluating and quantifying the effects of forecasting errors; and
(c)   evaluating the implications of low forecasting quality for a market mechanism.

The prediction task was fitted to the setup of a blockchain-based LEM. Thereby, the present research distinguishes itself notably from previous studies that solely try to forecast smart meter time series in general. The evaluation of forecasting errors and their implications was based on the commonly used market mechanism for discrete interval, double-sided auctions, while the forecasting error settlement structure was based on the work of Mengelkamp et al. [6]. The following research questions were examined:

1.   Which prediction technique yields the best 15-min ahead forecast for smart meter time series measured in 3-min intervals using only input features generated from the historical values of the time series and calendar-based features?
2.   Assuming a forecasting error settlement structure, what is the quantified loss of households participating in the LEM due to forecasting errors by the prediction technique identified in Question (a)?
3.   Depending on Question (b), what implications and potential adjustments for an LEM market mechanism can be identified?

The present research found that regressing with a least absolute shrinkage and selection operator (LASSO) on one week of historical consumption data is the most suitable approach to household-level energy forecasting. However, this method's forecasting errors still substantially diminish the economical benefit of a blockchain-based LEM. Thus, we conclude that changes to the market designs are the most promising way to still employ blockchain-based LEMs as means to meet some of the challenges generated by Germany's current energy transition.

The remainder of the paper is structured as follows: Section 2 presents the forecasting models and error measures used to evaluate the prediction accuracy. Moreover, it introduces the market mechanism and simulation used to evaluate the effect of prediction errors in LEMs. Section 3 describes the data used. Section 4 presents the prediction results of the forecasting models, evaluates their performance relative to a baseline model and assesses the effect of prediction errors on market outcomes. The insights gained from this are then used to identify potential adjustments for future market mechanisms. Finally, Section 5 concludes with a summary, limitations, and an outlook on further research questions that emerge from the findings of the present research.

All code and data used in the present research are available through the Quantnet website (www.quantlet.de). They can be easily found by entering BLEM (Blockchain-based Local Energy

Markets) into the search bar. As part of the Collaborative Research Center, the Center for Applied Statistics and Economics and the International Research Training Group (IRTG) 1792 at the Humboldt-University Berlin, Quantnet contributes to the goal of strengthening and improving empirical economic research in Germany.

## 2. Method

To select the forecasting technique, we applied the following criteria:

1.  The forecasting technique has to produce deterministic (i.e., point) forecasts.
2.  The forecasting technique had—for comparison—to be used in previous studies.
3.  The previous study or studies using the forecasting technique had to use comparable data, i.e., recorded by smart meters in 60-min intervals or higher resolution, recorded in multiple households, and not recorded in small and medium enterprises (SMEs) or other business or public buildings.
4.  The forecasting task had to be comparable to the forecasting task of the present research, i.e., single consumer household (in contrast to the prediction of aggregated energy time series) and very short forecasting horizon ($\leq$ 24 h).
5.  The forecasting technique had to take historical and calendar features only as input for the prediction.
6.  The forecasting technique had to produce absolutely and relative to other studies promisingly accurate predictions.

Based on these criteria, two forecasting techniques were selected for the prediction task at hand. As short-term energy forecasting techniques are commonly categorized into statistical and machine learning (or artificial intelligence) methods [25–27], one method of each category was chosen: Long short-term memory recurrent neural network (LSTM RNN) adapted from the procedure outlined by Shi et al. [23] and autoregressive LASSO as implemented by Li et al. [24]. Instead of LSTM RNN, gated recurrent unit (GRU) neural networks could have been used as well. However, despite needing fewer computational resources, their representational power may be lower compared to LSTM RNNs [28] and their successful applicability in household-level energy forecasting has not been proven in previous studies. The forecasting techniques used data from 1 January 2017 to 30 September 2017 as training input and the forecast was evaluated on data from 1 October 2017 to 31 December 2017. This means that no data from autumn were included in the training data. However, this seems unlikely to influence the forecasting performance as the German climate in the months from February to April (which are included in the training data) is comparable to the climate in the months from October to December; the forecasting horizon is very short-term; and the input for the forecasting techniques is too short to reflect any seasonal changes in temperature or sunshine hours.

### 2.1. Baseline Model

A frequent baseline model used for deterministic forecasts is the simple persistence model [29]. This model assumes that the conditions at time $t$ persist at least up to the period of forecasting interest at time $t + h$. The persistence model is defined as

$$\widehat{x}_{t+1} = x_t. \tag{1}$$

There are several other baseline models commonly used in energy load forecasting. Most of them are, in contrast to the persistence model, more sophisticated benchmarks. However, as the forecasting task at hand serves the specific use case of being an input for the bidding process in a blockchain-based LEM, the superiority of the forecasting model over a benchmark model is of secondary importance. Hence, in the present research, only the persistence model served as a baseline for the forecasting techniques presented in Sections 2.2 and 2.3.

### 2.2. Machine Learning-Based Forecasting Approach

The first sophisticated forecasting technique that was employed in the present research to produce as accurate as possible predictions for the blockchain-based LEM is a machine learning algorithm. Long short-term memory (LSTM) recurrent neural networks (RNN) have been introduced only very recently in load forecasting studies (e.g., [22,23,27,30]).

Neural networks do not need any strong assumptions about their functional form, such as traditional time series models (e.g., autoregressive moving average, ARMA). However, they are universal approximators for finite input [31] and, therefore, are especially well suited for the prediction of volatile time series such as energy consumption or production. The most basic building blocks of any neural network are three types of layers: an input layer, one or more hidden layer(s), and an output layer. Each layer consists of one or more units (sometimes called neurons). Each unit in a layer takes in an input, applies a transformation to this input, and outputs it to the next layer. Formally, this can be written as

$$
\begin{aligned}
\boldsymbol{h}_{1,i} &= \phi_1 \left( \boldsymbol{W}_1 \boldsymbol{x}_i + \boldsymbol{b}_1 \right) \\
\boldsymbol{h}_{2,i} &= \phi_2 \left( \boldsymbol{W}_2 \boldsymbol{h}_{1,i} + \boldsymbol{b}_2 \right) \\
&\vdots \\
o_i &= \phi_n \left( \boldsymbol{W}_n \boldsymbol{h}_{(n-1),i} + \boldsymbol{b}_n \right) = \widehat{y}_i,
\end{aligned}
\tag{2}
$$

where $n$ denotes a layer, $\phi_n$ is the activation function, $\boldsymbol{W}_n$ is the weight matrix, and $\boldsymbol{b}_n$ is the bias vector in layer $n$. $\boldsymbol{x}_i$ is the $i$th input vector and $o_i$ is the output value of the output layer, which is the estimation of the true value $y_i$. The weight matrices and bias vectors in each layer are parameters that are adjusted during the training of the model.

However, such a simple neural network is not particularly well-suited for time series learning [28]. This is because simple neural networks, such as the one described above, do not have an internal state that could retain a memory of previously processed input. That is, to learn a sequence or time series, the described neural network would always need the complete time series as a single input. It cannot retain a memory of something learned in a previous chunk of the time series to apply it to the next chunk that is fed into the model. This problem is tackled by recurrent neural networks.

RNNs still consist of the basic building blocks of units and layers. However, the units not only feed forward the transformed input as output but also have a recurrent connection that feeds an internal state back into the unit as input. Thereby, a RNN unit loops over individual elements of an input sequence, instead of processing the whole sequence in a single step. This means that the RNN unit applies the transformation to the first element of the input sequence and combines it with its internal state. This introduces the notion of time into neural networks. Formally, this can be written as

$$
\begin{aligned}
\boldsymbol{h}_{1,t} &= \phi_1 \left( \boldsymbol{W}_1^{(i)} \boldsymbol{x}_t + \boldsymbol{W}_1^{(r)} \boldsymbol{h}_{1,(t-1)} + \boldsymbol{b}_1 \right) \\
\boldsymbol{h}_{2,t} &= \phi_2 \left( \boldsymbol{W}_2^{(i)} \boldsymbol{h}_{1,t} + \boldsymbol{W}_2^{(r)} \boldsymbol{h}_{2,(t-1)} + \boldsymbol{b}_2 \right) \\
&\vdots \\
o_t &= \phi_n \left( \boldsymbol{W}_n^{(i)} \boldsymbol{h}_{(n-1),t} + \boldsymbol{b}_n \right) = \widehat{y}_t,
\end{aligned}
\tag{3}
$$

where $n$ denotes a layer, $\phi_n$ is the activation function, $\boldsymbol{W}_n^{(i)}$ is the weight matrix for the input, $\boldsymbol{W}_n^{(r)}$ is the weight matrix for the recurrent input (i.e., the output of layer $n$ in the previous time step), and $\boldsymbol{b}_n$ is the bias vector in layer $n$. $\boldsymbol{x}_t$ is the input vector at time $t$ and $o_t$ is the output value of the output layer which is the estimation of the true value $y_t$. Note that the output layer has no recurrent units but is the same as in a simple feed forward network.

The cyclical structure of an RNN unit can be unrolled across time (see Figure 1). This illustrates that a RNN is basically a simple neural network that has one layer for each time step that has to be processed per input. Theoretically, this feedback structure enables RNNs to retain information about sequence elements that have been processed many steps before the current step and use it for the prediction of the current step. However, in practice, the vanishing gradient problem occurs (for more details on the vanishing gradient problem, see, e.g., [32]). This problem makes RNNs basically untrainable for very long sequences.

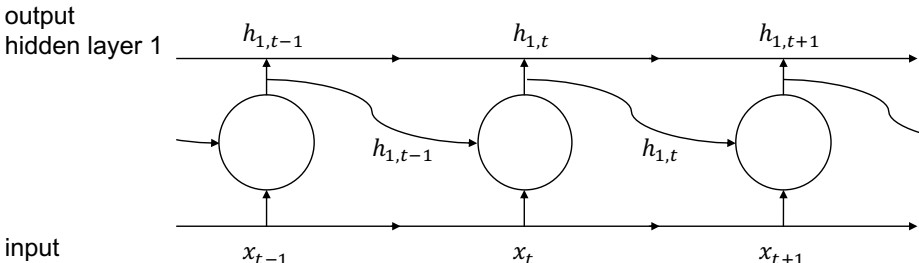

**Figure 1.** Schematic representation of an unfolded RNN unit. Adapted from [28].

To overcome the vanishing gradient problem, Hochreiter and Schmidhuber [33] developed LSTM units. LSTM RNN is an advanced architecture of RNN that is particularly well suited to learn long sequences or time series due to its ability to retain information over many time steps [28]. LSTM units extend RNN units by an additional state. This state can retain information for as long as needed. In which step this additional state is updated and in which state the information it retains is used in the transformation of the input is controlled by three so-called gates [34]. These three gates have the form of a simple RNN cell. Formally, by slightly adapting the notation of Lipton et al. [35]—who used $h_{t-1}$ instead of $s_{t-1}$, whereas the notation used here ($s_{t-1}$) accounts for the modern LSTM architecture with peephole connections—the gates can be written as

$$
\begin{aligned}
i_t &= \sigma\left(W^{(ix)}x_t + W^{(is)}s_{t-1} + b_i\right) \\
f_t &= \sigma\left(W^{(fx)}x_t + W^{(fs)}s_{t-1} + b_f\right) \\
o_t &= \sigma\left(W^{(ox)}x_t + W^{(os)}s_{t-1} + b_o\right),
\end{aligned}
\tag{4}
$$

where $\sigma$ is the sigmoid activation function $\sigma(z) = \frac{1}{1+e^{-z}}$, $W$ denotes the weight matrices that are intuitively labeled (*ix* for the weight matrix of gate $i_t$ multiplied with the input $x_t$ etc.), and $b$ denotes the bias vectors. Again, following the notation of Lipton et al. [35], the full algorithm of a LSTM unit is given by the three gates specified above, the input node,

$$
g_t = \sigma\left(W^{(gx)}x_t + W^{(gh)}h_{t-1} + b_g\right),
\tag{5}
$$

the internal state of the LSTM unit at time step $t$,

$$
s_t = g_t \odot i_t + s_{t-1} \odot f_t,
\tag{6}
$$

where $\odot$ is pointwise multiplication, and the output at time step $t$,

$$
h_t = \phi\left(s_t\right) \odot o_t.
\tag{7}
$$

The internal structure of a LSTM cell is further clarified in Figure 2. For an intuitive but more detailed explanation of LSTM neural networks, see [28] (Ch. 6.2).

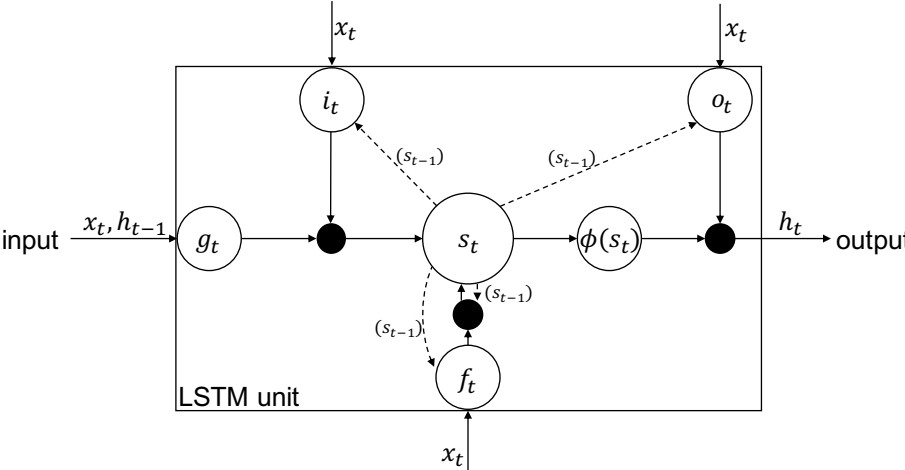

**Figure 2.** Schematic representation of an LSTM unit. Adapted from [36]. The filled in circles represent the pointwise multiplication operation denoted by $\odot$ in Equations (6) and (7).

In summary, LSTM RNNs are capable of learning highly complex, non-linear relationships in time series data, which makes them a promising forecasting technique to predict households' very short-term energy consumption and production.

The specific LTSM RNN approach adopted in the present research was based on the procedure employed by Shi et al. [23] to forecast individual households' energy consumption. According to the relevant use case in the present research, LSTM RNNs were trained for each household individually using only the household's historic consumption patterns and calendar features. Specifically, seven days of past consumption, an indicator for weekends, and an indicator for Germany-wide holidays were used as input for the neural network in the present research. This follows the one-hot encoding used by Chen et al. [30]. Seven days of lagged data were used as input because preliminary results indicated that the autocorrelation in the time series becomes very weak in lags beyond one week. Moreover, using the previous week as input data still preserves the weekly seasonality and represents a reasonable compromise between as much input as possible and the computational resources needed to process the input in the training process of the LSTM neural network. The target values in the model training were single consumption values in 15-min aggregation. The following example serves as illustration: Assume the consumption values in 3-min intervals from 13 November 2017 13:00 to 20 November 2017 13:00 and zero/one-indicators for weekends and holidays (i.e., $3 \times 3360$ data points) are fed into the neural network. The model then produces a single output value that estimates the household's energy consumption in kWh from 20 November 2017 13:00 to 20 November 2017 13:15.

A neural network is steered by several hyperparameters: the number and type of layers, the number of hidden units within each layer, the activation functions used within each unit, dropout rates for the recurrent transformation, and dropout rates for the transformation of the input. To identify a well working combination of hyperparameter values, tuning is necessary which is unfortunately computationally very resource intensive. Table 1 presents the hyperparameters that were tuned and their respective value ranges. The tuning was done individually for each network layer. Optimally, the hyperparameters of all layers should be tuned simultaneously. However, due to computational constraints, that was not possible here and, thus, the described, second-best option was chosen. As the hyperparameter values specified in Table 1 for layer 1 alone result in 81 possible hyperparameter combinations, only random samples of these combinations were taken, the resulting models trained on a randomly chosen dataset and compared. In total, 16 models with one layer, 13 models with two layers and 13 models with three layers were tuned. The model tuning was conducted on four Tesla P100 graphical processing units (GPUs) through the Machine Learning (ML) Engine of the Google Cloud Platform. The job was submitted to the Google Cloud ML Engine via Google Cloud SDK and the R package cloudml. Although neural networks can be trained much faster on GPUs than

on conventional central processing units (CPUs) [28], usage of GPUs through the Google Cloud ML Engine incurs substantial monetary cost. Thus, they were only used for the model tuning in this study.

**Table 1.** The hyperparameters that were tuned for an optimal LSTM RNN model specification.

| | Hyperparameter | Possible Values | Possible Combinations | Sampling Rate | # of Assessed Combinations |
|---|---|---|---|---|---|
| layer 1 | batch size hidden units recurrent dropout dropout | {128, 64, 32} {128, 64, 32} {0, 0.2, 0.4} {0, 0.2, 0.4} | 81 | 0.2 | 16 |
| layer 2 | hidden units recurrent dropout dropout | {128, 64, 32} {0, 0.2, 0.4} {0, 0.2, 0.4} | 26 | 0.5 | 13 |
| layer 3 | hidden units recurrent dropout dropout | {128, 64, 32} {0, 0.2, 0.4} {0, 0.2, 0.4} | 26 | 0.5 | 13 |

Based on the hyperparameter tuning results, a model with the specification shown in Table 2 was used for the prediction of a single energy consumption value for the next 15 min.

The total length of data points covered in the training process equals the batch size times the input data points times the number of data points that are aggregated for each prediction (i.e., 5 data points): $700 \times 32 \times 5 = 112,000$ data points. This is equivalent to the time period from 1 January 2017 00:00 to 22 August 2017 09:03. The tuning process and results can be replicated by following the Quantlet link in the caption of Table 2.

**Table 2.** Tuned hyperparameters for LSTRM RNN prediction model. **Q** BLEMtuneLSTM (github.com /QuantLet/BLEM/tree/master/BLEMtuneLSTM)

| Hyperparameter | Tuned Value |
|---|---|
| layers | 1 |
| hidden units | 32 |
| dropout rate | 0 |
| recurrent dropout rate | 0 |
| batch size | 32 |
| number of input data points | 3360 |
| number of training samples | 700 |
| number of validation samples | 96 |

The general procedure of model training, model assessment and prediction generation is shown in Procedure 1. The parameter tuple was set globally for all household datasets based on the hyperparameter tuning. Thereafter, the same procedure was repeated for each dataset: First, the consumption data time series was loaded, target values were generated, and the input data were transformed. The transformation consisted of normalizing the log-values of the consumption per 3-min interval between 0 an 1. This ensured fast convergence of the model training process. The data batches for the model training and the cross-validation were served to the training algorithm by so-called generator functions. Second, the LSTM RNN was compiled and trained with Keras, which is a neural network application programming interface (API) written in Python. The Keras R package (v2.2.0.9), which was used with RStudio v1.1.453 and TensorFlow 1.11.0 as back-end, is a wrapper of the Python library and is maintained by Chollet et al. [37]. The model training and prediction for each household was performed on a Windows Server 2012 with 12 cores and 24 logical processors of Intel Xeon 3.4 GHz CPUs. The model training was done in a differing number of epochs as early stopping was employed to prevent overfitting: Once the mean absolute error on the validation data did not decrease by more than 0.001 in three consecutive epochs, the training process was stopped. Third, the trained model was used to generate predictions on the test set that comprised data from 1 October 2017 00:00 to 1 January 2018 00:00 (i.e., 44,180 data points). As the prediction was made

in 15-min intervals, in total, 8836 data points were predicted. Using the error measures described in Section 2.4, the model performance was assessed. Finally, the predictions for all datasets were saved for the evaluation in the LEM market mechanism.

---

**Procedure 1** Supervised training of and prediction with LSTM RNN.

---

1: Set parameter tuple $< l, u, b, d >$: number of layers $l \subseteq L$, number of hidden LSTM-units $u \subseteq U$, batch size $b \subseteq B$, and dropout rate $d \subseteq D$.
2: Initiate prediction matrix $P$ and list for error measures $\Theta$.
3: **for** Household $i$ in dataset pool $I$ **do**
4:      Load dataset $\Psi_i$.
5:      Generate target values $y$ by aggregating data to 15-min intervals.
6:      Transform time series in dataset $\Psi_i$ and add calendar features.
7:      Set up training and validation data generators according to parameter tuple $< b, d >$.
8:      Split dataset $\Psi_i$ into training dataset $\Psi_{i,tr}$ and testing dataset $\Psi_{i,ts}$.
9:      Build LSTM RNN $\zeta_i$ on Tensorflow with network size $(l, h)$.
10:      **repeat**
11:         **At** $k$th epoch **do**:
12:         Train LSTM RNN $\zeta_i$ with data batches $\varphi_{train} \subseteq \Psi_{i,tr}$ supplied by training data generator.
13:         Evaluate performance with mean absolute error $\Lambda_k$ on cross-validation data batches $\varphi_{val} \subseteq \Psi_{i,tr}$ supplied by validation data generator.
14:      **until** $\Lambda_{k-1} - \Lambda_k < 0.001$ for the last 3 epochs.
15:      Save trained LSTM RNN $\zeta_i$.
16:      Set up testing data generator according to tuple $< b, d >$.
17:      Generate predictions $\hat{y}_i$ with batches $\varphi_{ts} \subseteq \Psi_{i,ts}$ fed by testing data generator into LSTM RNN $\zeta_i$.
18:      Calculate error measures $\Theta_i$ to assess performance of $X_i$.
19:      Write prediction vector $\hat{y}_i$ into column $i$ of matrix $P$.
20: **end for**.
21: Save matrix $P$.
22: **End.**

---

### 2.3. Statistical Method-Based Forecasting Approach

To complement the machine learning approach of a LSTM RNN with a statistical approach, a second, regression-based method was used. For this purpose, the autoregressive LASSO approach proposed by Li et al. [24] seemed most suitable. Statistical methods have the advantage of much lower model complexity compared to neural networks which makes them computationally much less resource intensive.

Li et al. [24] used LASSO [38] to find a sparse autoregressive model that generalizes better to new data. Formally, the LASSO estimator can be written as

$$\widehat{\beta}_{\text{LASSO}} = \underset{\beta}{\arg\min} \frac{1}{2} \left\| (y - X\beta) \right\|_2^2 + \lambda \left\| \beta \right\|_1, \tag{8}$$

where $X$ is a matrix with row $t$ being $[1 \quad x_t^T]$ (the length of $x_t^T$ is the number of lag-orders $n$ available), and $\lambda$ is a parameter that controls the level of sparsity in the model, i.e., which of the $n$ available lag-orders are included to predict $y_{t+1}$. This model specification selects the best recurrent pattern in the energy time series by shrinking coefficients of irrelevant lag-orders to zero and, thereby, improves the generalizability of the prediction model. In the present research, the sparse autoregressive LASSO approach was implemented using the R package glmnet [39]. As for the LSTM RNN approach, model training and prediction were performed for every household individually. Following the procedure of Li et al. [24] , only historical consumption values were used as predictors. Specifically, for comparability to the LSTM approach, seven days of lagged consumption values served as input to the LASSO model. The response vector consisted of single consumption values in 15-min aggregation. The same example as above serves as illustration: Assume the consumption values in 3-min intervals from 13 November 2017 13:00 to 20 November 2017 13:00 (i.e., 3360 data points) are available to the model for prediction. Based on the training data, the model chooses the lagged values with the highest predictive power

and makes a linear estimation of a single value for the household's energy consumption in kWh from 20 November 2017 13:00 to 20 November 2017 13:15.

The detailed description of the model estimation and prediction is presented in Procedure 2. As the LASSO model requires a predictor matrix, the time series of each household was split in sequences of length $n = 3360$ with five data points skipped in between. The skip accounted for the fact that the response vector was comprised of 15-min interval consumption values (i.e., five aggregated 3-min values). After generating the predictor matrix for the model estimation, the optimal $\lambda$ was found in a $K$-fold cross-validation. Here, $K$ was set to 10. The sequence of $\lambda$-values that was tested via cross-validation was of length $L = 100$ and was constructed by calculating the minimum $\lambda$-value as a fraction of the maximum $\lambda$-value ($\lambda_{min} = \varepsilon\lambda_{max}$, where $\lambda_{max}$ was such that all $\beta$-coefficients were set equal to zero) and moving along the log-scale from $\lambda_{max}$ to $\lambda_{min}$ in $L$ steps. However, the glmnet algorithm used early-stopping to reduce computing times if the percent of null deviance explained by the model with a certain $\lambda$ did not change sufficiently from one to the next $\lambda$-value. The cross-validation procedure identified the biggest $\lambda$ that is still within one standard deviation of the $\lambda$ with the lowest mean absolute error. The final coefficients for each household were then computed by solving Equation (8) for the complete predictor matrix. Thereafter, the predictions were made on the testing data. Again, the time series was sliced according to the sliding window of length $n = 3360$ skipping five data points and written into a predictor matrix. This matrix comprised data from 1 October 2017 00:00 to 1 January 2018 00:00 (i.e., 8836 cases of 3360 lagged values), resulting again in 8836 predicted values as in the case of the LSTM approach. The predictions on all datasets were assessed using the error measures described in Section 2.4 and saved for the evaluation of the prediction in the context of the LEM market mechanism.

---

**Procedure 2** Cross-validated selection of $\lambda$ for LASSO and prediction.

1: Initiate prediction matrix $P$ and list for error measures $\Theta$.
2: **for** Household $i$ in dataset pool $I$ **do**
3:    Load dataset $\Psi_i$.
4:    Generate target values $\boldsymbol{y}$ by aggregating data to 15-min intervals.
5:    Split dataset $\Psi_i$ into training dataset $\Psi_{i,tr}$ and testing dataset $\Psi_{i,ts}$.
6:    Generate predictor matrix $M_{tr}$ by slicing time series $\Psi_{i,tr}$ with sliding window.
7:    Generate sequence of $\lambda$-values $\{l_s\}_{s=1}^{L}$.
8:    Set number of cross-validation (CV) folds $K$.
9:    Split predictor matrix $M_{tr}$ into $K$ folds.
10:     **for** $k$ in $K$ **do**
11:        Select fold $k$ as CV testing set and folds $j \neq k$ as CV training set.
12:        **for** each $l_s$ in $\{l_s\}_{s=1}^{L}$ **do**
13:            Compute vector $\widehat{\boldsymbol{\beta}}_{k,l_s}$ on CV training set.
14:            Compute mean absolute error $\Lambda_{k,l_s}$ on CV testing set.
15:        **end for**.
16:     **end for**.
17:     For each $\widehat{\boldsymbol{\beta}}_{k,l_s}$ calculate average mean absolute error $\bar{\Lambda}_s$ across the $K$ folds.
18:     Select cross-validated $\lambda$-value $l_s^{CV}$ with the highest regularization (min no. of non-zero $\beta$-coeff.) within one SD of the minimum $\bar{\Lambda}_s$.
19:     Compute $\widehat{\boldsymbol{\beta}}_{l_s^{CV}}$ on complete predictor matrix $M_{tr}$.
20:     Generate predictor matrix $M_{ts}$ by slicing time series $\Psi_{i,ts}$ with sliding window.
21:     Generate predictions $\widehat{\boldsymbol{y}}_i$ from predictor matrix $M_{ts}$ and coefficients $\widehat{\boldsymbol{\beta}}_{l_s^{CV}}$.
22:     Calculate error measures $\Theta_i$ to assess performance.
23:     Write prediction vector $\widehat{\boldsymbol{y}}_i$ into column $i$ of matrix $P$.
24: **end for**.
25: Save matrix $P$.
26: **End.**

### 2.4. Error Measures

Forecasting impreciseness is measured by a variety of norms. The $L_1$-type mean absolute error (MAE) is defined as the average of the absolute differences between the predicted and true values [40]:

$$\text{MAE} = \frac{1}{N} \sum_{t=1}^{N} |\widehat{x}_t - x_t|, \tag{9}$$

where $N$ is the length of the forecasted time series, $\widehat{x}_t$ is the forecasted value and $x_t$ is the observed value. As MAE is only a valid error measure if one can assume that for the forecasted distribution the mean is equal to the median (which might be too restrictive), an alternative is the root mean square error (RMSE), i.e., the square root of the average squared differences [29,41]:

$$\text{RMSE} = \sqrt{\frac{1}{N} \sum_{t=1}^{N} (\widehat{x}_t - x_t)^2}. \tag{10}$$

Absolute error measures are not scale independent, which makes them unsuitable to compare the prediction accuracy of a forecasting model across different time series. Therefore, they are complemented with the percentage error measures mean absolute percentage error (MAPE) and normalized root mean square error (NRMSE) normalized by the true value:

$$\text{MAPE} = \frac{100}{N} \sum_{t=1}^{N} \left| \frac{\widehat{x}_t - x_t}{x_t} \right|, \tag{11}$$

and

$$\text{NRMSE} = \sqrt{\frac{100}{N} \sum_{t=1}^{N} \left( \frac{\widehat{x}_t - x_t}{x_t} \right)^2}. \tag{12}$$

However, as Hyndman and Koehler [42] pointed out, using $x_t$ as denominator may be problematic as the fraction $\frac{\widehat{x}_i - x_i}{x_i}$ is not defined for $x_t = 0$. Therefore, time series containing zero values cannot be assessed with this definition of the MAPE and NRSME.

To overcome the shortage of an undefined fraction in the presence of zero values in the case of MAPE and NRMSE, the mean absolute scaled error (MASE) as proposed by Hyndman and Koehler [42] was used. That is, MAE was normalized with the in-sample mean absolute error of the persistence model forecast:

$$\text{MASE} = \frac{\text{MAE}}{\frac{1}{n-1} \sum_{t=2}^{N} |x_t - x_{t-1}|}. \tag{13}$$

In summary, in the present research, the forecasting performance of the LSTM RNN and the LASSO were evaluated using MAE, RMSE, MAPE, NRMSE, and MASE.

### 2.5. Market Simulation

We used a market mechanism with discrete closing times in 15-min intervals. Each consumer and each prosumer submit one order per interval and the asks and bids are matched in a closed double auction that yields a single equilibrium price. The market mechanism was implemented in R. This allows for a flexible and time-efficient analysis of the market outcomes with and without prediction errors.

The simulation of the market mechanism followed five major steps: First, the consumption and production values of each market participant per 15-min interval from 1 October 2017 00:00 to 1 January 2018 00:00 were retrieved. These values are either the true values as yielded by the aggregation of the raw data or the prediction values as estimated by the best performing prediction model. Second, for each market participant, a zero-intelligence limit price was generated by drawing randomly from

the discrete uniform distribution $U\{12.31, 24.69\}$. The lower bound is the German feed-in tariff of 12.31 $\frac{\text{EURct}}{\text{kWh}}$ and the upper bound is the average German electricity price in 2016 of 28.69 $\frac{\text{EURct}}{\text{kWh}}$ [43]. This agent behavior has been shown to generate efficient market outcomes in double auctions [44] and is rational in so far as electricity sellers would not accept a price below the feed-in tariff and electricity buyers would not pay more than the energy utility's price per kWh. However, this assumes that the agents do not consider any non-price related preferences, such as strongly preferring local renewable energy [6]. Third, for each trading slot (i.e., every 15-min interval), the bids and asks were ordered in price-time precedence. Given the total supply is lower than the total demand, the lowest bid price that can still be served determines the equilibrium price. Given the total supply is higher than the total demand, the overall lowest bid price determines the equilibrium price. In the case of over- or undersupply, the residual amounts are traded at the feed-in (12.31 $\frac{\text{EURct}}{\text{kWh}}$) or the regular household consumer electricity tariff (28.69 $\frac{\text{EURct}}{\text{kWh}}$) with the energy utility. Fourth, the applicable price for each bid and ask was determined and the settlement amounts, resulting from this price and the energy amount ordered, were calculated. In the case of using predicted values for the bids, there was an additional fifth step: After the next trading period, when the actual energy readings were known, any deviations between predictions and true values were settled with the energy utility using the feed-in or household consumer electricity tariff. This led to correction amounts that were deducted or added to the original settlement amounts. For the market simulation, perfect grid efficiency and, hence, no transmission losses were assumed.

## 3. Data

The raw data used for the present research were provided by Discovergy GmbH and are available at  BLEMdata (github.com/QuantLet/BLEM/tree/master/data), hosted at GitHub. Discovergy describes itself as a full-range supplier of smart metering solutions offering transparent energy consumption and production data for private and commercial clients [45]. To be able to offer such data-driven services, Discovergy smart meters record energy consumption and production near real-time—i.e., in 2-s intervals—and send the readings to Discovergy's servers for storage and analysis. Therefore, Discovergy has extremely high resolution energy data of their customers at their disposal. This high resolution is in stark contrast to the half-hourly or even hourly recorded data used in previous studies on household energy forecasting (e.g., [21,23,46,47]). To our knowledge, there is no previous research using Discovergy smart meter data, apart from Teixeira et al. [48], who used the data as simulation input but not for analysis or prediction.

The data come in 200 individual datasets each containing the meter readings of a single smart meter; 100 datasets belong to pure energy consumers and 100 datasets belong to energy prosumers (households that produce and consumer energy). The meter readings were aggregated to 3-min intervals and range from 1 January 2017 00:00 to 1 January 2018 00:00. This translated into 175,201 observations per dataset. Each observation consists of the total cumulative energy consumption and the total cumulative energy production from the date of installation until time $t$, current power over all phases installed in the meter at time $t$ and a timestamp in Unix milliseconds.

For further analysis, the power readings were dropped and the first differences of the energy consumption and production readings were calculated. These first differences are equivalent to the energy consumption and production within each 3-min interval between two meter recordings. The result of this computation leaves each dataset with two time series (energy consumption and energy production in kWh) and 175,200 observations.

Figure 3 shows the energy consumption time series of Consumer 082. In the first panel of Figure 3, the consumption per 3-min interval for all of 2017 is shown. Notably, there are two extended periods (in March and June) and three shorter periods (in July, September, and December) with a clearly distinguishable low consumption level and low fluctuation. The most likely explanations for these low, stable energy consumption periods are holidays, in which the household members are on vacation and leave appliances that are on standby or always turned on as the only energy consumers.

The second panel zooms to just one month making daily fluctuation patterns visible. The last panel zooms in to a single day of energy consumption. It exemplifies well a usual pattern of energy consumption: There is low and rather stable energy consumption from midnight until about 07:30, which only fluctuates in a systematic and repeated way due appliances in standby and "always on" appliances, such as a fridge and/or freezer. At around 07:30, the household members probably wake up and the energy consumption spikes for the next 30 min—the lights are turned on, coffee is made, the stove is turned on, and maybe a flow heater is used to shower with hot water. As the household members leave the house (13 May is a Monday), the consumption slowly decreases again. In the evening at about 18:30 the energy consumption spikes again, probably caused by dinner preparations.

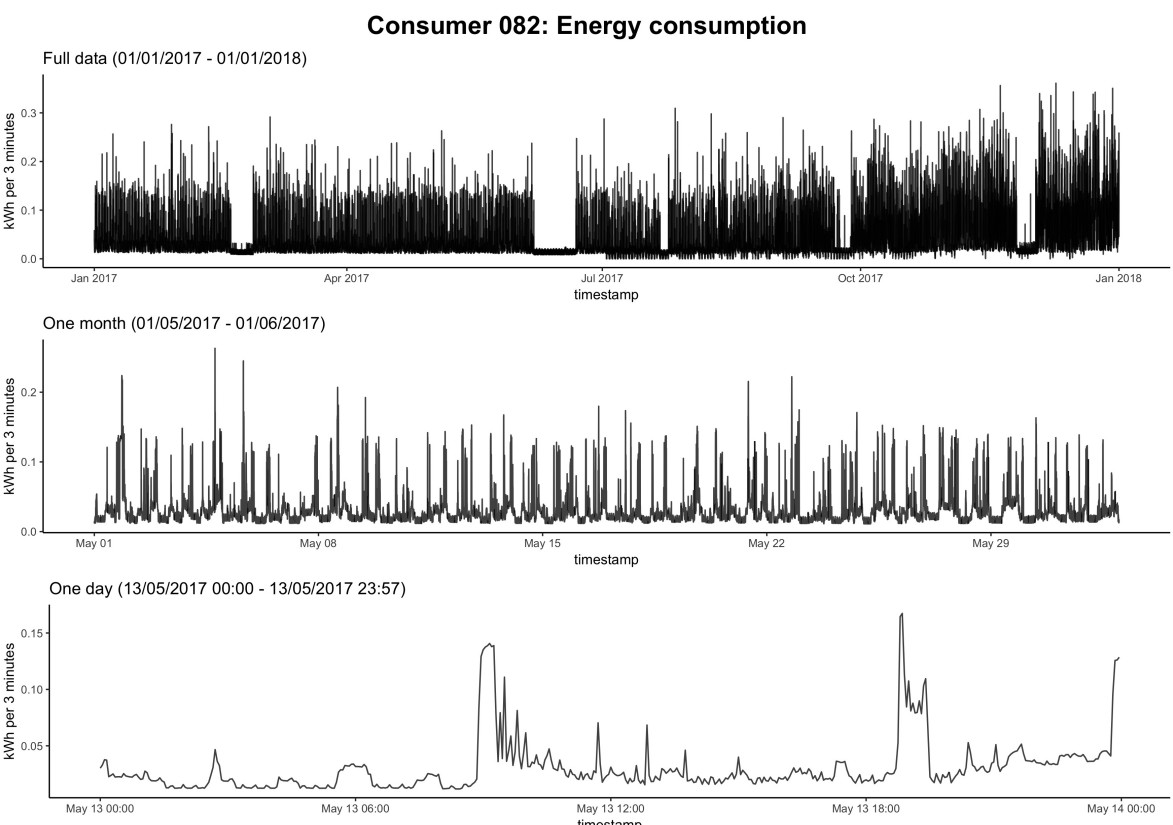

**Figure 3.** Energy consumption recordings of Consumer 082. The first panel shows the full year 2017, the second panel zooms in to one month (May), and the third panel zooms in to one day (13 May). BLEMplotEnergyData (github.com/QuantLet/BLEM/tree/master/BLEMplotEnergyData)

Out of the 100 consumer datasets, five exhibited non-negligible shares of zero consumption values leading to their exclusion. One consumer dataset was excluded as the consumption time series was flat for the most part of 2017 and one consumer was excluded due to very low and stable consumption values with very rare, extreme spikes. Four more consumers were excluded due to conspicuous regularity in daily or weekly consumption patterns. Lastly, one consumer was excluded not due to peculiarities in the consumption patterns but due to missing data. As the inclusion of this shorter time series would have led to difficulties in the forecasting algorithms, this dataset was excluded as well.

Out of the 100 prosumer datasets, 86 were excluded due to zero total net energy production in 2017. These "prosumers" would not act as prosumers in an LEM as they would never actually supply a production surplus to the market. Of the remaining 14 prosumer datasets, one prosumer dataset was excluded because the total net energy it fed into the grid in 2017 was just 22 kWh. Additionally,

one prosumer dataset was excluded as it only fed energy into the grid in the period from 6 January 2017 to 19 January 2017. For all other measurement points, the net energy production was zero.

Overall, 88 consumer and 12 prosumer datasets remained for the analysis. All datasets include a timestamp and the consumption time series for consumers and the production time series for prosumers with a total of 175,200 data points each.

## 4. Results

### 4.1. Evaluation of the Prediction Models

Three prediction methods were used to forecast the energy consumption of 88 consumer households 15 min ahead: a baseline model, a LSTM RNN model, and a LASSO regression. All three prediction models were compared and evaluated using the error measures presented in Section 2.4. The performance of the prediction models was tested on a quarter of the available data. That is, the prediction models were fitted on the consumption values from 1 January 2017 00:00 to 30 September 2017 00:00, which is equivalent to 131,040 data points per dataset. For all 88 consumer datasets, the models were fitted separately resulting in as many distinct LASSO and LSTM prediction models. The fitted models were then used to make energy consumption predictions in 15-min intervals for each household individually on the data from 1 October 2017 00:00 to 1 January 2018 00:00. This equates to 8836 predicted values per dataset per prediction method.

Figure 4 displays the total sum of over- and underestimation errors in kWh of each prediction method per dataset. That is, for each consumer, the total sum of overestimation errors is calculated as summing all differences between true and forecasted value, when the forecasted value is greater than the true value (formally, $\delta_i^o = \sum_{t=1}^{N} (\hat{x}_{i,t} - x_{i,t}) [(\hat{x}_{i,t} - x_{i,t}) > 0]$; red bars) and the total sum of underestimation errors is calculated as summing all differences between true and forecasted value, when the forecasted value is smaller than the true value (formally, $\delta_i^u = \sum_{t=1}^{N} (\hat{x}_{i,t} - x_{i,t}) [(\hat{x}_{i,t} - x_{i,t}) < 0]$; blue bars). Thus, the red and blue bars added together depict the total sum of errors in kWh for each prediction method per dataset.

The LASSO technique achieved overall lower total sums of errors than the baseline model. Notably, the sum of underestimation errors is higher across the datasets than the sum of overestimation errors. This points towards a general tendency of underestimating sudden increases in energy consumption by the LASSO technique. The LSTM model on the other hand shows a much higher variability in the sums of over- and underestimation errors. By tendency, the overestimation errors of the LSTM model are smaller than those of the LASSO and baseline model. Nevertheless, the underestimation is much more pronounced in the case of the LSTM model. Especially, some datasets stand out regarding the high sum of underestimation errors. This points towards a much higher heterogeneity in the suitability of the LSTM model to predict consumption values depending on the energy consumption pattern of the specific dataset. The LASSO technique on the other hand seems to be more equally well suited for all datasets and their particular consumption patterns.

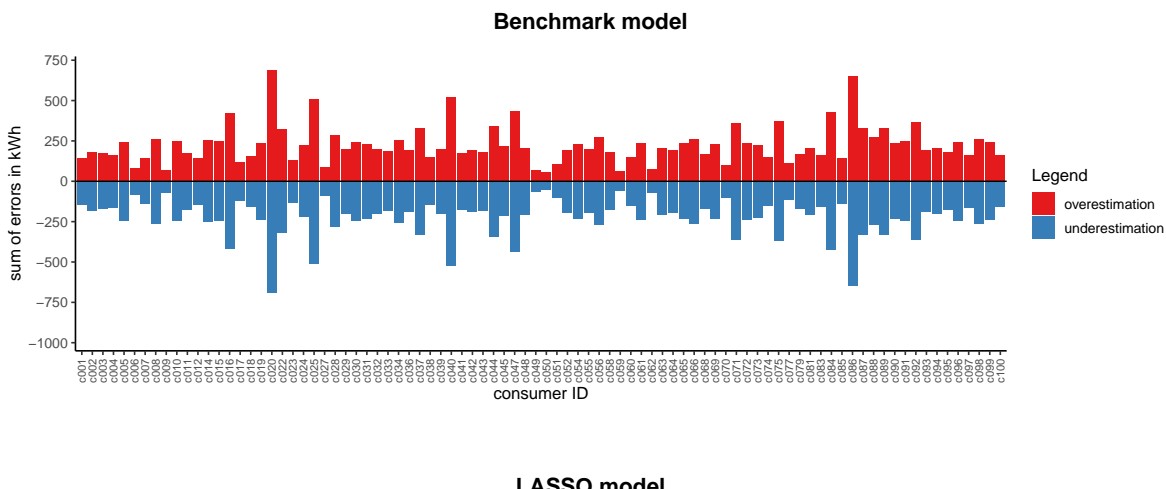

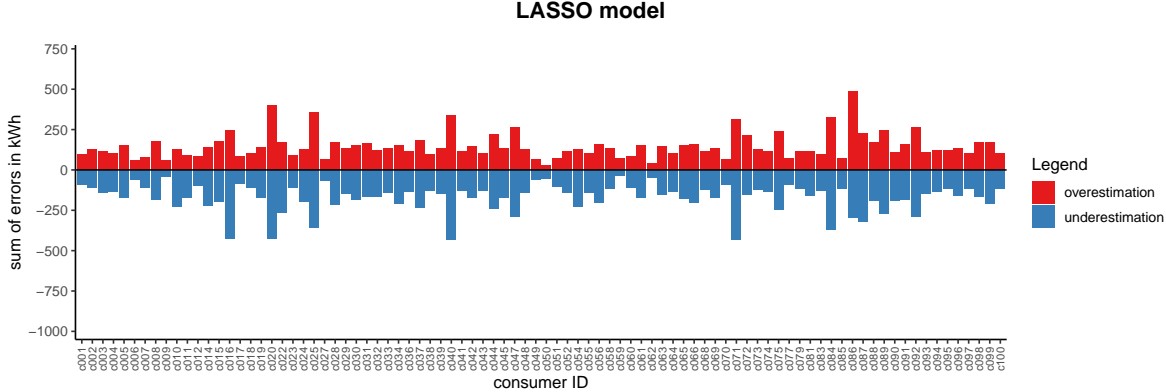

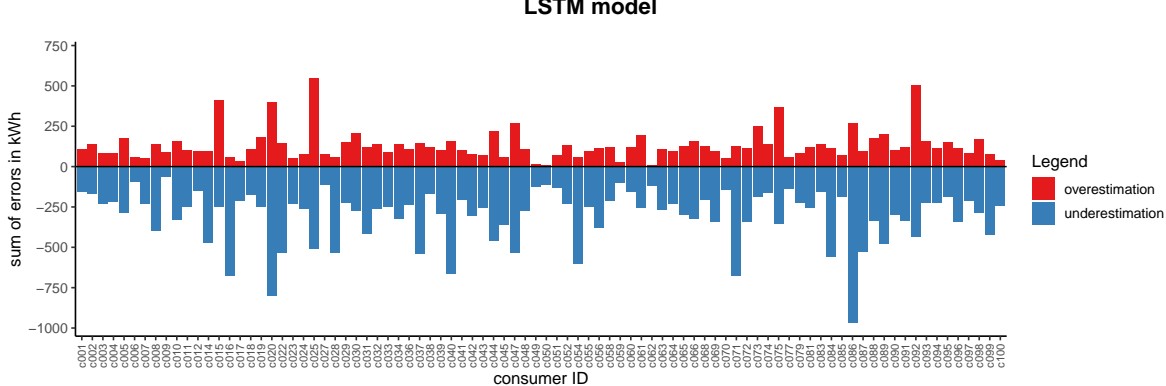

**Figure 4.** Sum of total over- and underestimation errors of energy consumption per consumer dataset and prediction model. ◯ BLEMplotPredErrors (github.com/QuantLet/BLEM/tree/master/BLEMplotPredErrors)

The average performance of the three prediction models across all 88 datasets is shown in Table 3. As can be seen, LASSO and LSTM consistently outperformed the baseline model according to MAE, RMSE, MAPE, NRMSE and MASE. The LASSO model performed best overall with the lowest median error measure scores across the 88 consumer datasets.

**Table 3.** Median of error measures for the prediction of energy consumption across all 88 consumer datasets. ◯ BLEMevaluateEnergyPreds (github.com/QuantLet/BLEM/tree/master/BLEMevaluateEnergyPreds)

| Model | MAE | RMSE | MAPE | NRMSE | MASE |
|---|---|---|---|---|---|
| LSTM | 0.04 | 0.09 | 22.22 | 3.30 | 0.85 |
| LASSO | 0.03 | 0.05 | 17.38 | 2.31 | 0.57 |
| Benchmark | 0.05 | 0.10 | 27.98 | 5.08 | 1.00 |
| Improvement LSTM (in %) | 16.21 | 12.61 | 20.57 | 34.98 | 14.78 |
| Improvement LASSO (in %) | 44.02 | 48.73 | 37.88 | 54.61 | 43.02 |

The superior performance of the LASSO model is also clearly visible in Figure 5. This might be surprising, as from a theoretical point of view, a linear model should not outperform a non-linear neural network that fulfills the conditions for a universal approximator for finite input. The most reasonable explanation seems to be that the LSTM RNN model used here missed a good local minimum for a number of datasets and converged to suboptimal parameter combinations. If the main focus of this paper were finding an optimal forecasting algorithm for individual households' short-term energy consumption, this would require further investigation. However, this study focused on the achievable forecasting accuracy with state-of-the-art methods already employed in previous studies. The results imply that it seems unwise to use a general set of hyperparameters on a number of household energy consumption datasets that differ quite substantially in their energy consumption patterns. However, as the LASSO technique employed here achieved an error score that is competitive with comparable research applications, the underperformance of the LSTM RNN compared to the LASSO technique is of no further concern.

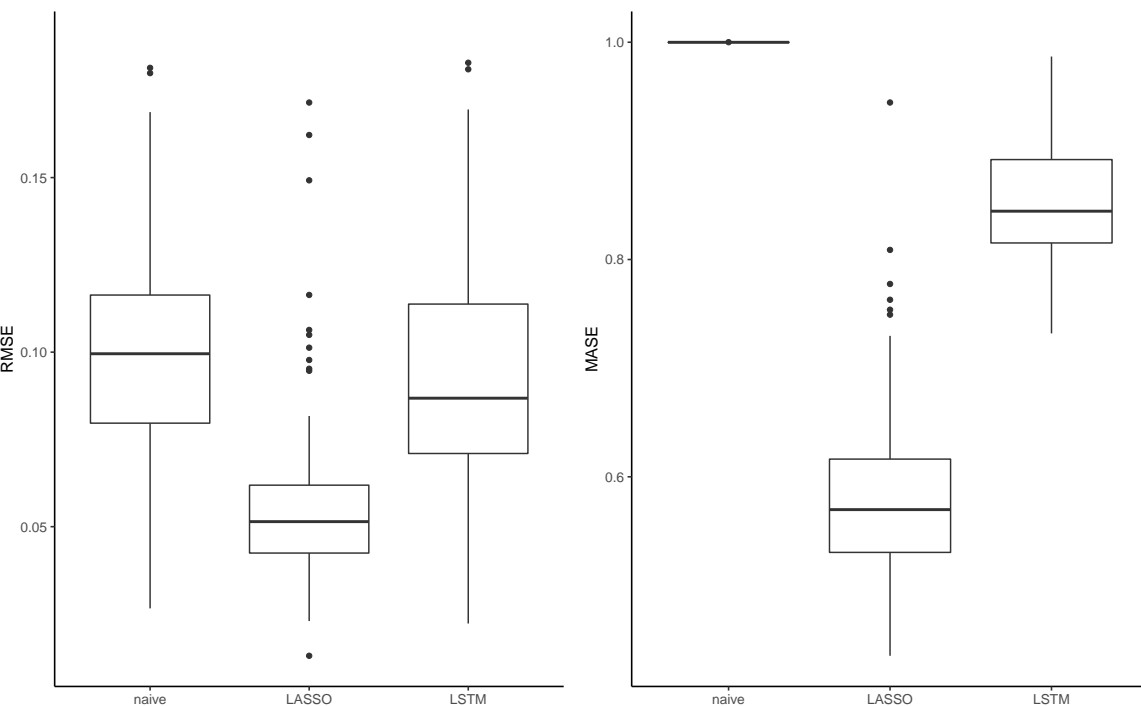

**Figure 5.** Box plots of RMSE and MASE scores across 88 consumer datasets for the three different prediction models (the upper 3%-quantile of the error measures is cut off for better readability). ◯ BLEMevaluateEnergyPreds (github.com/QuantLet/BLEM/tree/master/BLEMevaluateEnergyPreds)

Interestingly, some consumer datasets exhibit apparently much harder to predict consumption patterns than the other datasets. This is exemplified by the outliers of the box plots, as well as by the heat map displayed in Figure 6. It confirms that there is quite some variation among the same prediction methods across different households. Therefore, one may conclude that there is no "golden industry standard" approach for households' very short-term energy consumption forecasting. Nevertheless, it is obvious that the LASSO model performed best overall. Hence, the predictions on the last quarter of the data produced by the fitted LASSO model for each consumer dataset were used for the evaluation of the market simulation presented next.

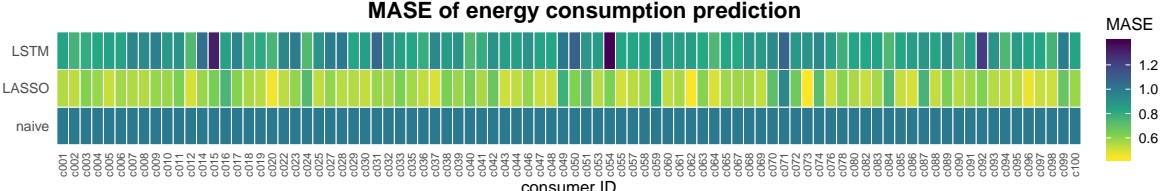

**Figure 6.** Heat map of MASE scores for the prediction of consumption values per consumer dataset. BLEMevaluateEnergyPreds (github.com/QuantLet/BLEM/tree/master/BLEMevaluateEnergyP reds)

## 4.2. Evaluation of the Market Simulation

The market simulation used the market mechanism of a discrete interval, closed double auction to assess the impact of prediction errors on market outcomes. In total, 88 consumers and 12 prosumer datasets were available. To evaluate different supply scenarios, the market simulation was conducted three times with a varying number of prosumers included. The three scenarios consisted of a market simulation with balanced energy supply and demand, a simulation with severe oversupply and a simulation with severe undersupply. To avoid extreme and unusual market outcomes over the time period of the simulation, two prosumers with high production levels, but long periods of no energy production in the simulation period were not included as energy suppliers in the market. The remaining prosumers were in- or excluded according to the desired supply scenario. That is, the undersupply scenario comprised six prosumers, the balanced supply scenario additionally included one more, and the oversupply scenario included additionally to the balanced supply scenario two more prosumers.

### 4.2.1. Market Outcomes in Different Supply Scenarios

The difference between supply and demand for each trading period, the equilibrium price of each double auction, and the weighted average price—termed LEM price—is shown in Figure 7. The LEM price is computed in each trading period as the average of the auctions equilibrium price and the energy utilities energy price (28.69 $\frac{EURct}{kWh}$) weighted by the amount of kWh traded for the respective price. The three graphs below depicting the market outcomes are results of the market simulation with true consumption values.

As can be seen, the equilibrium price shown in the middle panel of Figure 7 moves roughly synchronous to the over-/undersupply shown in the top panel. As there is by tendency more undersupply in the balanced scenario (the red line in the top panel indicates perfectly balanced supply and demand), the equilibrium price is in most trading periods close to its upper limit and the LEM price is almost always above the equilibrium price. There is by tendency more undersupply due to the fact that four of the relevant prosumer datasets are from producers with large capacities (>10 kWh per 15-min interval) that dominated the remaining prosumers' production capacity substantially and therefore a more balanced supply scenario could not be created.

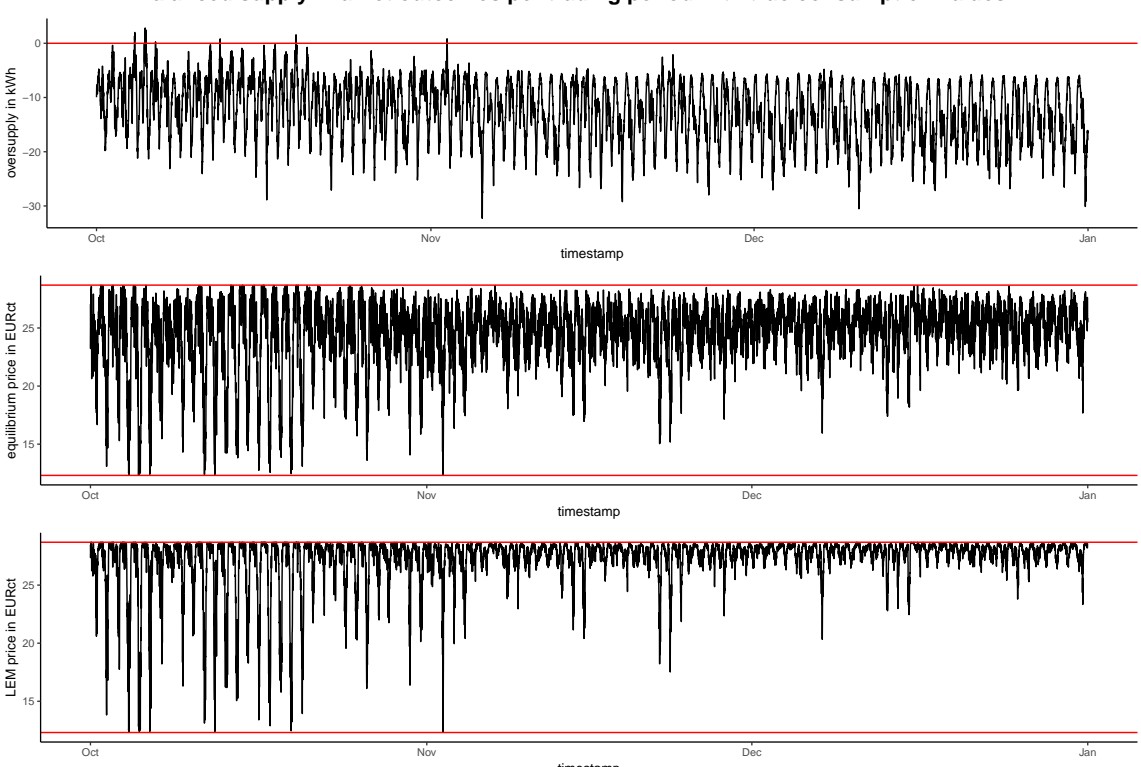

**Figure 7.** Market outcomes per trading period simulated with true values and a balanced supply scenario. ⬤ BLEMmarketSimulation (github.com/QuantLet/BLEM/tree/master/BLEMmarketSimulation)

This observation is in contrast to the oversupply scenario shown in Figure 8. Here, the prosumers' energy supply surpasses the consumers' energy demand in the majority of trading periods. Accordingly, the equilibrium price in each auction is close to the lower limit of the energy utility's feed-in tariff of 12.31 $\frac{\text{EURct}}{\text{kWh}}$. However, trading periods with undersupply lead to visible spikes in the equilibrium price, which are, as expected, even more pronounced in the LEM price. In all other periods, the equilibrium price equals the LEM price as all demand is served by the prosumers and there is no energy purchased from the grid.

Figure 9 shows the market simulation performed in an undersupply scenario. Here, the market outcomes are the opposite to the oversupply scenario: The equilibrium prices move in a band between 20 $\frac{\text{EURct}}{\text{kWh}}$ and the upper limit of 28.69 $\frac{\text{EURct}}{\text{kWh}}$. The LEM prices are even higher as the deficit in supply has to be compensated by energy purchases from the grid. This means that, the more severe the undersupply is, the more energy has to be purchased from the grid, and the more the LEM price surpasses the equilibrium price.

In summary, one can conclude that the market outcomes are the more favorable to consumers, the more locally produced energy is offered by prosumers. Assuming a closed double auction as market mechanism and zero-intelligence bidding behavior of market participants, oversupply reduces the LEM prices substantially leading to savings on the consumer side. On the other hand, prosumers will favor undersupply in the market as they profit from the high equilibrium prices while still being able to sell their surplus energy generation at the feed-in tariff without a loss compared to no LEM.

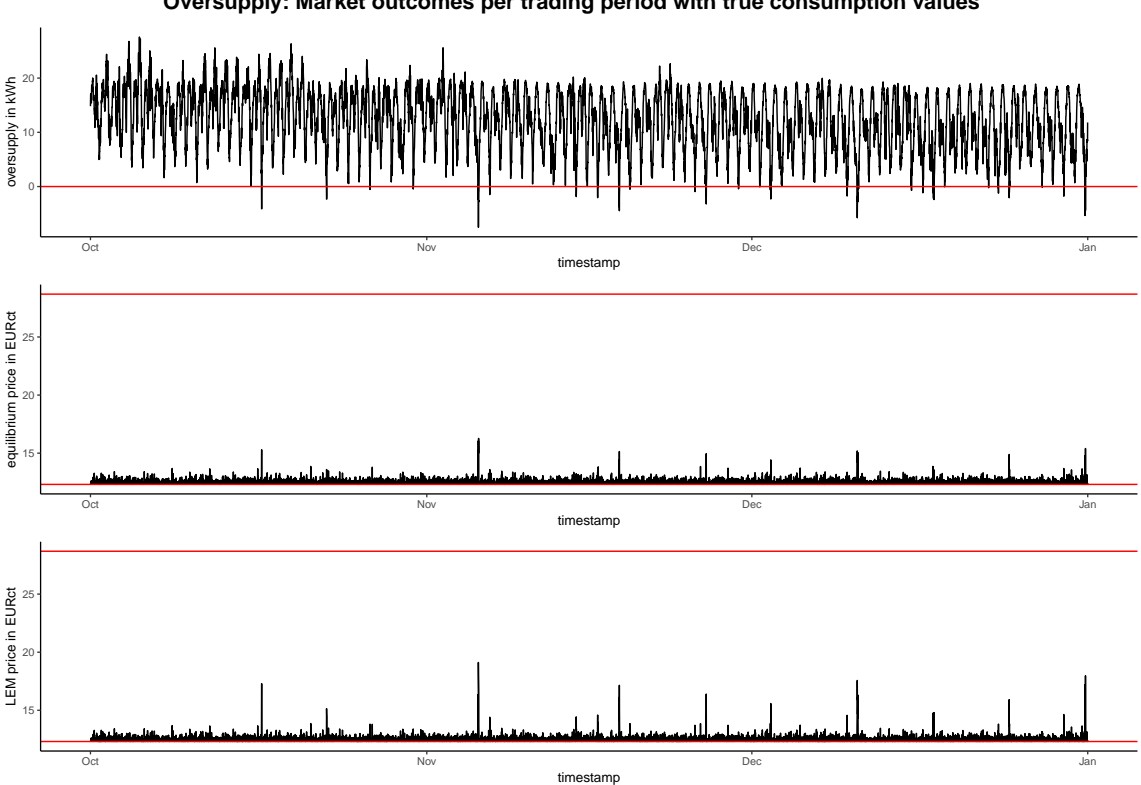

**Figure 8.** Market outcomes per trading period simulated with true values and an oversupply scenario.
 BLEMmarketSimulation (github.com/QuantLet/BLEM/tree/master/BLEMmarketSimulation)

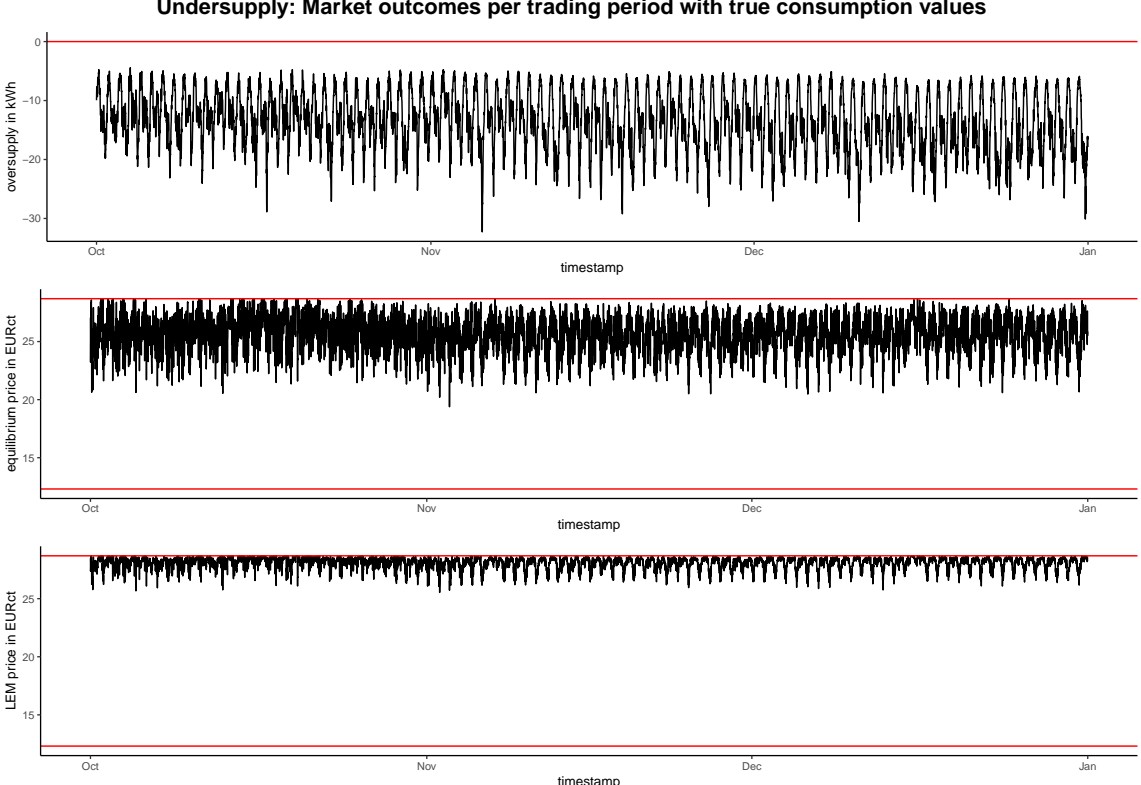

**Figure 9.** Market outcomes per trading period simulated with true values and an undersupply scenario.
 BLEMmarketSimulation (github.com/QuantLet/BLEM/tree/master/BLEMmarketSimulation)

### 4.2.2. Loss to Consumers due to Prediction Errors

To assess the adverse effect of prediction errors on market outcomes, the LASSO-predicted energy consumption values per 15-min interval are used. The predictions of the model served as order amounts in the auction bids. After the true consumption in the respective trading period was observed, payments to settle over- or underestimation errors were made. That is, if a consumer bid with a higher amount than actually consumed, it still bought the full bid amount from the prosumers but had to sell the surplus to the energy utility over the grid at the feed-in tariff. On the other hand, if a consumer bid with a lower amount than actually consumed, it bought the bid amount from the prosumers but had to purchase the surplus energy consumption from the grid at the energy utility's tariff. Thus, prediction errors are costly as the consumer always has to clear the order in less favorable conditions than the equilibrium price provides.

Table 4 contrasts the results of the market simulation with true consumption values with the results of the market simulation with predicted consumption values in three different supply scenarios. The equilibrium and LEM prices almost do not differ within the three scenarios whether the true or predicted consumption values are used. The prices between the scenarios, however, differ substantially. The average total revenue over the three-month simulation period of prosumers is largely unaffected by the use of true or predicted consumption values. This is not surprising as the revenue is a function of the equilibrium price, which is apparently largely unaffected by whether true or predicted consumption values are used, and the electricity produced, which is obviously completely unaffected by whether true or predicted consumption values are used.

**Table 4.** Average results of the market simulation for three different supply scenarios. Prices are averaged across all trading periods. Revenues and costs for the whole simulation period are averaged across all prosumers and consumers, respectively. ◉ BLEMevaluateMarketSim (github.com/QuantLe t/BLEM/tree/master/BLEMevaluateMarketSim)

| Mean | Balanced Supply | | Oversupply | | Undersupply | |
|---|---|---|---|---|---|---|
| | True | Predicted | True | Predicted | True | Predicted |
| Equilibrium price (in EURct) | 24.64 | 24.61 | 12.50 | 12.49 | 25.68 | 25.69 |
| LEM price (in EURct) | 27.31 | 27.28 | 12.51 | 12.49 | 28.08 | 28.10 |
| Revenue (in EUR) | 1113.84 | 1108.88 | 3454.62 | 3451.69 | 1035.90 | 1036.12 |
| Cost with LEM (in EUR) | 439.26 | 457.94 | 200.75 | 226.61 | 451.60 | 470.69 |
| Cost without LEM (in EUR) | 459.83 | 446.93 | 459.83 | 446.93 | 459.83 | 446.93 |

What differs according to Table 4, however, is the cost for consumers. The cost without the LEM is on average across all consumers smaller when using predicted consumption values compared to using true consumption values. This can be explained by the LASSO model's tendency to underestimate on the data at hand and because correction payments for the prediction errors are not factored into this number. The average total cost for electricity consumption in the whole simulation period is with an LEM higher when using predicted consumption values compared to using true consumption values. This is due to the above-mentioned need to settle prediction errors at unfavorable terms.

The percentage loss induced by prediction errors is shown in Table 5. Depending on the supply scenario it ranges between about 4.8% and 13.75%. These numbers have to be judged relative to the savings that are brought to consumers by the participation in an LEM. It turns out, that in the balanced supply scenario, the savings due to the LEM are almost completely offset by the loss due to prediction errors. As consumers profit more from an LEM, the lower the equilibrium prices are, this is not the case in the oversupply scenario. Here, the savings are substantial and amount to about 130%, which is almost ten times more than the percentage loss due to the prediction errors. However, the problem of the settlement structure for prediction errors becomes very apparent in the undersupply scenario. Here, the savings due to an LEM are more than offset by the loss due to prediction errors. Consequently, consumers would be better off not participating in an LEM.

**Table 5.** Average savings for consumers due to the LEM and average loss for consumers due to prediction errors in the LEM. 🔍 BLEMevaluateMarketSim (github.com/QuantLet/BLEM/tree/master /BLEMevaluateMarketSim)

| Mean | Balanced Supply | Oversupply | Undersupply |
|------|----------------:|-----------:|------------:|
| Cost without LEM (in EUR) | 459.83 | 459.83 | 459.83 |
| Cost predicted values (in EUR) | 457.94 | 226.61 | 470.69 |
| Cost true values (in EUR) | 439.26 | 200.75 | 451.60 |
| Savings due to LEM (in %) | 4.82 | 129.08 | 1.90 |
| Loss due to pred. errors (in %) | −4.80 | −13.75 | −4.76 |

This result is visualized in a more differentiated way in Figure 10. The figure shows for each supply scenario, for each consumer, the total energy cost over the whole simulation period in: (1) no LEM; (2) an LEM with the use of predicted consumption values; and (3) an LEM with the use of true consumption values. For each supply scenario, the bottom panel shows the percentage loss due to not participating in the LEM and the loss due to participating and using predicted consumption values compared to participating and using true consumption values. In the balanced scenario, there are some consumers who would make a loss due to the participation in the LEM and relying on predicted values.

For them, the loss due to no LEM (yellow bar) is smaller than the loss due to prediction errors (green bar). However, 56 out of 88 consumer (i.e., 64%) also profit from the participation in the LEM despite the costs induced by prediction errors. Due to the much lower equilibrium prices in the oversupply scenario, the LEM participation here is, despite prediction errors, profitable for all consumers. However, even in this scenario, the savings for the consumers are diminished by more than 10%, which is quite substantial. In contrast, in the undersupply scenario, the loss due to the prediction errors leaves the participation in the LEM for almost all consumers unprofitable. Merely three consumers would profit and have lower costs in an LEM than without an LEM, despite prediction errors.

Overall, it becomes clear that prediction errors significantly lower the economic profitability of an LEM for consumers. This, however, is often argued to be one of the main advantages of LEMs. The result is especially concerning in LEMs where locally produced energy is undersupplied. Here—still assuming the closed double auction market mechanism and zero-intelligence bidding strategies—the savings from the participation in the LEM are marginal. Therefore, the costs induced by prediction errors mostly outweigh the savings from the participation. This results in an overall loss for consumers due to the LEM, which makes the participation economically irrational. Only in cases of substantial oversupply, the much lower equilibrium price, compared to the energy utility's price, compensates for the costs from prediction errors.

In conclusion, this means that LEMs with a discrete interval, closed double auction as market mechanism and a prediction error settlement structure as proposed in [6] combined with the prediction accuracy of state-of-the-art energy forecasting techniques require substantial oversupply in the LEM for it to be beneficial to consumers.

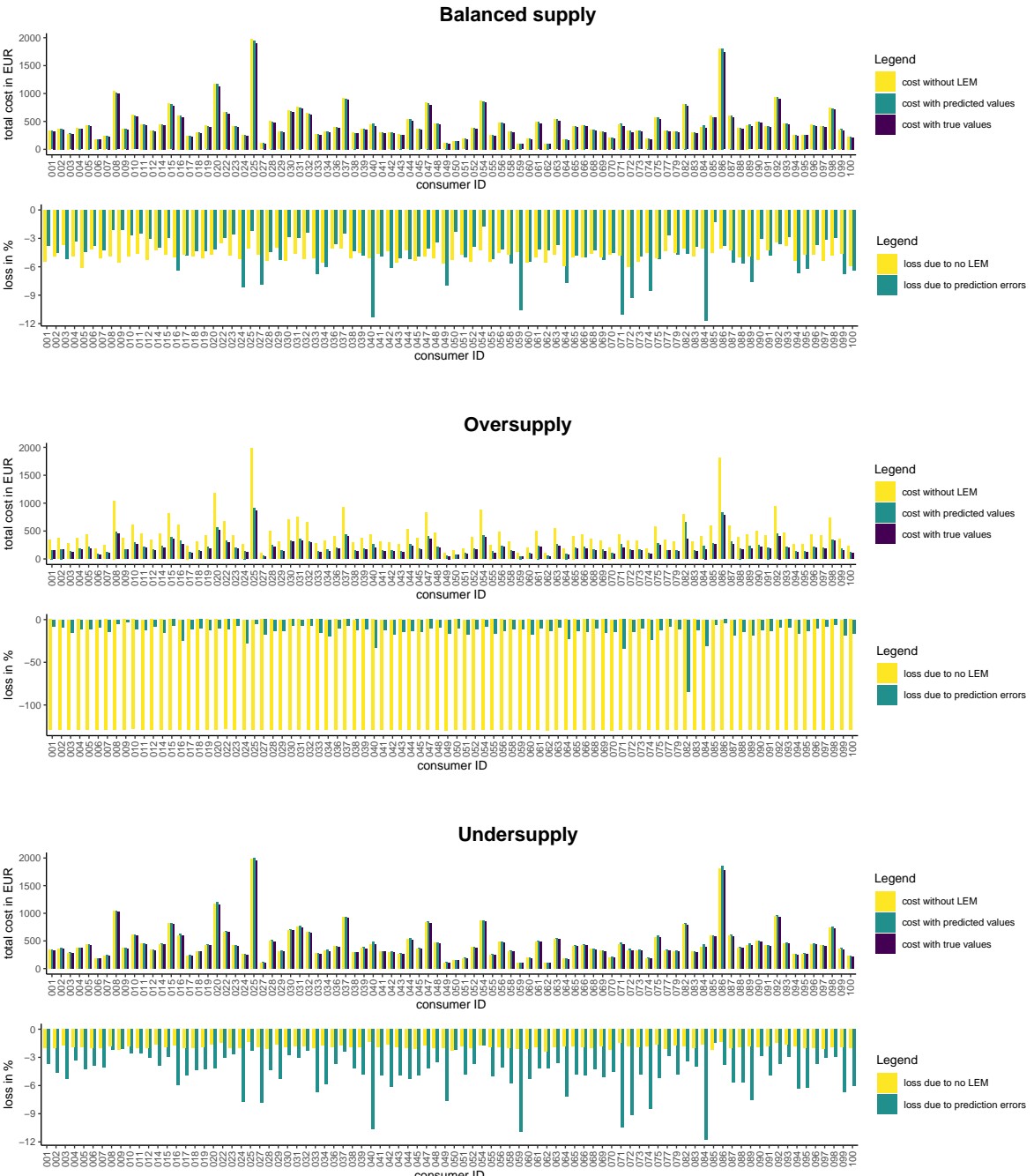

**Figure 10.** Total energy cost to consumers from 01 October 2018 to 31 December 2017 in the case of
no LEM, LEM with true values, and LEM with predicted values in three different supply scenarios.
BLEMevaluateMarketSim (github.com/QuantLet/BLEM/tree/master/BLEMevaluateMarketSim)

### 4.3. Implications for Blockchain-Based Local Energy Markets

In light of these results, it remains open to derive implications and to propose potential
adjustments for an LEM market mechanism. After all, there are substantial advantages of LEMs
which have been established in various studies and still make LEMs an attractive solution for the
challenges brought about by the current energy transition. Adjustments mitigating the negative effect
of prediction errors on the profitability of LEMs could address one or more of the following areas: first,
the forecasting techniques employed; second, the demand and supply structure of the LEM; and, third,
the market mechanism used in the blockchain-based LEM.

The first and most intuitive option is to improve the forecasting accuracy with which the predictions, which serve as the basis of bids and asks, are made. For example, a common approach to reduce the bias of LASSO-based predictions are post-LASSO techniques such as presented by Chernozhukov et al. [49]. Another aspect that seems relevant for the improvement of forecasting models is the evaluation method. Using economic measures for the evaluation of forecasting model performance may address a potential mismatch between statistical measures of forecasting accuracy and the resulting economic profits [50]. However, these approaches most likely result only in small improvements. Thus, the most obvious way to achieve a substantial improvement is the inclusion of more data. More data may hereby refer either to a higher resolution of recorded energy data or to a wider range of data sources such as behavioral data of household members or data from smart appliances. A higher resolution of smart meter readings is already easily achievable. The smart meters installed by Discovergy that also supplied the data for the present research are capable of recording energy measurements up to every two seconds. However, data at such a fine granularity requires substantial data storage and processing capacities which are unlikely to be available in an average household. Especially, the training of prediction models with such vast amounts of input data points is computationally very resource intensive. The potential solution of outsourcing this, however, introduces new data privacy concerns that are already a sensible topic in smart meter usage and blockchain-based LEMs (e.g., [8,51]). Increasing the forecasting intervals to 30 or 60 min, as an alternative way to reduce the computational resources needed, would presumably decrease the forecasting accuracy which, in turn, might increase the cost for consumers. However, the effect of this potential solution on the cost for consumers due to forecasting errors seems reasonable to be investigated in future studies. The inclusion of behavioral data into prediction models such as the location of the person within their house and the inclusion of smart appliances' energy consumption (as done by Kong et al. [22]) and running schedules raises important privacy concerns as well. Pooling and using energy consumption data of several households, as done by Shi et al. [23], again introduces privacy concerns as it implies data sharing between households, which in relatively small LEMs cannot be guaranteed to preserve the anonymity of market participants. For all these reasons, it seems unlikely that in the near future qualitative jumps in the prediction accuracy of very short-term household energy consumption or production of individual households will be available.

The second option addresses the demand and supply structure in the blockchain-based LEM. As shown in Section 4.2, the cost induced by prediction errors and their settlement is more than compensated in an oversupply scenario. Hence, employing LEMs only in a neighborhood in which energy production surpasses energy consumption would mitigate the problem of unprofitability due to prediction errors as well. Where this is not possible, participation to the LEM could be restricted, such that oversupply in a majority of trading periods is ensured. However, this might end up in a market manipulation that most likely makes most of LEMs' advantages obsolete. Moreover, it is unclear on what basis the restriction to participate in the market should be grounded.

The third option to mitigate the problem is the market mechanism and the prediction error settlement structure. A simple approach to reduce forecasting errors is to decrease the forecasting horizon. Thus, instead of having 15-min trading periods, which also require 15-min ahead forecast, the trading periods could be shrunk to just 3 min. This would increase the forecasting accuracy, and, thereby, lead to lower costs due to the settlement of prediction errors. However, in a blockchain-based LEM, more frequent market closings come at the cost of more computational resources needed for transaction verification and cryptographic block generation. Depending on the consensus mechanism used for the blockchain, the energy requirements for the computations, which secure transactions and generate new blocks, may be substantial. This, of course, is rather detrimental to the idea of promoting more sustainable energy generation and usage. Nevertheless, using consensus mechanisms based on identity verification of the participating agents may serve as a less computational, and thus energy intensive alternative, which might make shorter trading intervals reasonable. Another, more radical, approach might be to change the market mechanism of closed

double auctions altogether and use an exposed market instead. Hereby, the energy consumption and production is settled in an auction after the true values are known, instead of in advance. This means that market participants submit just limit prices in their bids and asks without related amounts and the offers are matched in an auction in regular time intervals. Then, the electricity actually consumed and produced in the preceding period is settled according to the market clearing price. Related to this approach is a solution, where bidding is based on forecasted energy values, while the settlement is shifted by one period such that the actual amounts can be used for clearing. This approach, however, may introduce the possibility of fraud and market manipulation as agents can try to deliberately bid using false amounts. While in the smart contracted developed by Mengelkamp et al. [6] funds needed to back up the bid are held as pledges until the contract is settled (this ensures the availability of the necessary funds to pay the bid), this would be senseless, if settlement is only based on actual consumption without considering the amount specified in the offer. However, the extent of this problem and ways to mitigate it should be assessed from a game theoretical perspective that is out of scope of the present research.

Overall, prediction errors have to be taken into account for future designs of blockchain-based LEMs. Otherwise, they may substantially lower the profitability and diminish the incentive to participate in an LEM for consumers. In addition, the psychological component of having to rely on an unreliable prediction algorithm that may be more or less accurate depending on the household's energy consumption patterns seems unattractive. Even though possible solutions are not trivial and each comes with certain trade-offs, there is room for future improvement of the smart contracts and the market mechanism they reproduce.

## 5. Conclusions

The present research had the objectives: (1) to evaluate the prediction accuracy achievable for household energy consumption with state-of-the-art forecasting techniques; (2) to assess the effect of prediction errors on an LEM that uses a closed double auction with discrete time intervals as market mechanism; and (3) to infer implications based on the results for the future design of blockchain-based LEMs.

In the performance assessment of currently used forecasting techniques, the LASSO model yielded the best results with an average MAPE across all consumer datasets of 17%. It was subsequently used to make predictions for the market simulation. The evaluation of the market mechanism and prediction error settlement structure revealed that, in a balanced supply and demand scenario, the costs of prediction errors almost completely offset savings brought by the participation in the LEM. In an undersupply scenario, the cost due to prediction errors even surpassed the savings and made market participation uneconomical. The most promising approach to mitigate this problem seemed to be adjustment of the market design, which can be two-fold: either shorter trading periods could be introduced, which would reduce the forecasting horizon, and therefore prediction errors, or the auction mechanism could be altered to not use predicted consumption values to settle transactions.

For the present research, data from a greater number of smart meters and more context information about the data would have been desirable. However, due to data protection legislation, no information regarding locality of the households, household characteristics or the type of power plant prosumer households used could be provided. Thus, unfortunately, no other covariates (e.g., temperature) could be used in the forecasting of energy consumption. In addition, the large-scale differences in the production capacities of the prosumers, contained in the data, complicated the analysis of the market simulation further. Additionally, it is worth mentioning that the market simulation did not account for taxes or fees, especially grid utilization fees, which can be a substantial share of the total electricity cost of households. The simulation also did not take into account compensation costs for blockchain miners that reimburses them for the computational cost they bear.

Evidently, future research concerned with blockchain-based LEMs should take into account the potential cost of prediction errors. Furthermore, to our knowledge, there has been no simulation

of a blockchain-based LEM with actual consumption and production data conducted. Doing so on a private blockchain with the market mechanism coded in a smart contract should be the next step for the assessment of potential technological and conceptual weaknesses.

In conclusion, previous research has shown that blockchain technology and smart contracts combined with renewable energy production can play an important role in tackling the challenges of climate change. The present research, however, emphasizes that advancement on this front cannot be made without a holistic approach that takes all components of blockchain-based LEMs into account. Simply assuming that reasonably accurate energy forecasts for individual households will be available once the technical challenges of implementing an LEM on a blockchain are solved, may steer research into a wrong direction and bears the risk of missing the opportunity to quickly move into the direction of a more sustainable and less carbon-intensive future.

**Author Contributions:** Conceptualization, M.K. and W.K.H.; Data curation, M.K.; Formal analysis, W.K.H.; Methodology, M.K.; Software, M.K.; Supervision, W.K.H.; Validation, M.K. and W.K.H.; Visualization, M.K.; Writing—original draft, M.K.; and Writing—review and editing, M.K. and W.K.H.

**Funding:** This research received no external funding.

**Acknowledgments:** We would like to thank Discovergy GmbH for the kind provision of their smart meter data, the Humboldt Lab for Empirical and Quantitative Research (LEQR) at the School of Business and Economics, Humboldt-University Berlin for the kind provision of computing resources, and the IRTG 1792 at the School of Business and Economics, Humboldt University of Berlin for valuable support.

**Conflicts of Interest:** The authors declare no conflict of interest.

**Data Availability:** All data and algorithms are freely available through Q www.quantlet.de with the keyword *BLEM* and at GitHub: github.com/QuantLet/BLEM.

## Abbreviations

The following abbreviations are used in this manuscript:

| | |
|---|---|
| LEM | Local energy market |
| LASSO | Least absolute shrinkage and selection operator |
| RNN | Recurrent neural network |
| LSTM | Long short-term memory |
| ML | Machine learning |
| GPU | Graphical processing unit |
| CPU | Central processing unit |
| CV | Cross-validation |
| SD | Standard deviation |
| MAE | Mean absolute error |
| RMSE | Root mean square error |
| MAPE | Mean absolute percentage error |
| NRMSE | Normalized root mean square error |
| MASE | Mean absolute scaled error |

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
