# Peer review of "Forecasting in Blockchain-Based Local Energy Markets"

_energies, doi:10.3390/en12142718_

Round 1

Reviewer 1 Report

- abbreviation should be defined before they are used
- explain in more details how recurrent neural networks and long short-term memory work (perhaps a figure helps)
- time to train the neural network (the use GPU should be explained)
- why a neural network has less performance than LASSO? this should be emphasized in the paper for the this specific application
- how accurate is market simulation ?

- the role of forecast should be better explained in context of blockchain  ( has the forecast the same importance in a non-blockchain solution or it is less important?)

Reviewer 2 Report

In this work the authors evaluate how two different forecasting techniques for individual households’ energy consumption affect blockchain-based local energy markets. Below are my comments/suggestions to the authors.

1. Lenzi et al., 2017, Dias et al., 2009 and Dias et al., 2013 propose methods to estimate consumer type-specific mean energy curves from aggregated data from multiple consumers. Would their work be relevant to a blockchain-based LEM? Could the authors explain the difference between their approach and the approach of Lenzi et al., 2017, Dias et al., 2009 and Dias et al., 2013?

2. On page 4 around line 132, I could not find the definition of SMEs.

3. Could the authors explain why other covariates (e.g., temperature) are not considered in the forecasting techniques?

4. Page 4 around line 168, the authors mention they adapted the notation of [33], but a better explanation would help readers as the relationship between h and s is still not clear. A bit more details on LSTM would greatly improve the text. 

5. Page 4 around line 148, would the terminology baseline model be more appropriate than benchmark model? The word benchmark is often used the to best method already available, which is not the case for the persistent model. It could be also called naïve model/approach.

6. Page 7 around line 252, how n=3,360 was obtained? It corresponds to data on every 3 minutes over a week (7 days), correct? But later you also talk about data aggregated on every 15 minutes. It would be great if these numbers could be better explained.

7. Section 4.1, since the training data was between 01.01.2017 to 30.09.2017 it did not include any data from fall/winter. How would this affect the predictions?

8. Figure 1, what kind of error was considered in this figure? Could you better explain how this figure was generated? Could you also add boxplots to facilitate comparison among methods?

9. Figure 2, there are 3 cases that LSTM led to very high MASE, any reason/intuition of why is that? Could the authors plot the time series for some of the consumers so the readers could see how those actually look like?

10. In the discussion the authors mention the problem of data storage and processing capacities in a household. Would it make sense to study the effects of increasing the time intervals from 15 to 30 minutes or even one hour on the final savings to consumers?

References

- Dias, R., Garcia, N., and Martarelli, A. (2009), “Non-parametric estimation for aggregated functional

data for electric load monitoring,” Environmetrics, 20, 111–130.

- Dias, R., Garcia, N. L., and Schmidt, A. M. (2013), “A hierarchical model for aggregated functional

data,” Technometrics, 55, 321–334.

- Lenzi, A., de Souza, C. P., Dias, R., Garcia, N. L., and Heckman, N. E. (2017), “Analysis of aggregated functional data from mixed populations with application to energy consumption,” Environmetrics, 28.
